# Provider Preference, Logistical Challenges, or Vaccine Hesitancy? Analyzing Parental Decision-Making in School Vaccination Programs: A Qualitative Study in Sydney, Australia

**DOI:** 10.3390/vaccines13010083

**Published:** 2025-01-17

**Authors:** Leigh McIndoe, Alexandra Young, Cristyn Davies, Cassandra Vujovich-Dunn, Stephanie Kean, Michelle Dives, Vicky Sheppeard

**Affiliations:** 1South Eastern Sydney Public Health Unit, Sydney, NSW 2031, Australia; 2Kirby Institute, University of New South Wales Sydney, Sydney, NSW 2052, Australia; ayoung@kirby.unsw.edu.au; 3Specialty of Child and Adolescent Health, Faculty of Medicine and Health, University of Sydney, Sydney, NSW 2050, Australia; cristyn.davies@sydney.edu.au; 4Sydney Infectious Diseases Institute, University of Sydney, Sydney, NSW 2050, Australia; 5School of Public Health, University of Queensland, Brisbane, QLD 4006, Australia; c.vujovichdunn@uq.edu.au; 6School of Public Health, University of Sydney, Sydney, NSW 2050, Australia

**Keywords:** parents, attitudes, beliefs, adolescent vaccination, school-based immunization, vaccine hesitancy, stakeholders, public health strategies

## Abstract

**Background:** School-based immunization programs are crucial for equitable vaccine coverage, yet their success depends on parental consent processes. This study investigates patterns of vaccine decision-making within Australia’s school-based immunization program, specifically focusing on human papillomavirus (HPV) and diphtheria-tetanus-pertussis (dTpa) vaccines offered free to adolescents aged 12–13. **Methods:** This qualitative study was conducted in the South Eastern Sydney Local Health District (2022–2023). Semi-structured interviews were held with school staff (*n* = 11) across government, Catholic, and independent schools, parents whose children were not vaccinated at school (*n* = 11) and a focus group with public health unit staff (*n* = 5). Data were analyzed to identify key barriers and patterns in vaccine decision-making. **Results:** Analysis revealed three distinct groups of parents whose children were not vaccinated through the school program: (1) those favoring general practitioners for vaccination, driven by trust in medical providers and a preference for personalized care; (2) those intending to consent but facing logistical barriers, including communication breakdowns and online consent challenges; and (3) vaccine-hesitant parents, particularly regarding HPV vaccination, influenced by safety concerns and misinformation. These findings demonstrate that non-participation in school vaccination programs should not be automatically equated with vaccine hesitancy. **Conclusions:** Tailored interventions are necessary for addressing vaccine non-participation. Recommendations include strengthening collaboration with general practitioners, streamlining consent processes and providing targeted education to counter misinformation. This study provides valuable insights into social determinants of vaccine acceptance and offers actionable strategies for improving vaccine uptake in school-based programs.

## 1. Introduction

Vaccination remains a cornerstone of national public health programs, playing a critical role in preventing the spread of infectious diseases worldwide. In Australia, the National Immunisation Program (NIP) offers government-funded essential vaccines to children, adolescents and adults, to ensure widespread community protection from a range of preventable diseases [1]. NIP vaccines include a single dose of the human papillomavirus (HPV) vaccine and a diphtheria-tetanus-pertussis (dTpa) booster which are offered to adolescents in Year 7 (ages 12–13 years) in high school. These vaccines are provided free of charge to students through the NIP, though a provider fee may apply outside of the school setting. The dTpa booster has been routinely offered in the school program since 2004, while the HPV vaccine was introduced as a 3-dose schedule for girls in year 7 in 2007 and extended to boys in 2013. A 2-dose HPV schedule was adopted in 2017 which continued until a single dose was considered fully vaccinated from 2023 [2,3,4,5]. Both vaccines are administered at the same visit, aiming to increase coverage and reduce the logistical burden for families and healthcare providers.

In the state of New South Wales (NSW), public health units (PHUs) are responsible for collaborating with schools to deliver school vaccination programs within their geographic areas. Almost all schools agree to host the immunization program and designate a school staff member to coordinate efforts and liaise with the PHU. In the South Eastern Sydney Local Health District (SESLHD), a team of nurses and support staff from the PHU deliver vaccination clinics to secondary schools across the district. Despite the SESLHD program’s overall success, with approximately 80% coverage for both HPV and dTpa vaccines, there is significant variation in vaccination rates between schools [5]. A recent post-COVID-19 pandemic decline in vaccine coverage has raised concerns regarding the program’s long-term effectiveness and equitable access [6], particularly in light of Australia’s goal to eliminate cervical cancer by achieving 90% HPV vaccination coverage by 2030 [7].

Obtaining parental or guardian consent presents a major challenge to achieving optimal vaccine coverage in school-based programs. Barriers include logistical challenges like incomplete or missed consent forms and absenteeism, as well as belief-based and perceived barriers such as personal or cultural beliefs, safety concerns and vaccine mistrust [8,9,10,11]. These factors are compounded by the global rise in vaccine hesitancy, which threatens the progress of immunization programs worldwide [12]. Understanding and addressing these barriers is critical for designing effective interventions to sustain vaccine uptake and confidence in a rapidly evolving public health landscape.

This qualitative study examines parental decision-making in SESLHD’s school vaccination program during a period of significant operational changes, including the transition to a single-dose HPV regimen and the introduction of an online parental consent platform. Conducted between 2022 and 2023, the research situates vaccine hesitancy within a broader context of logistical and systemic barriers, while also offering a timely analysis of the post-COVID-19 dynamics and their impact on vaccination behaviors [6].

The study builds on previous research on the SESLHD school vaccination program, focusing on barriers to consent and vaccine uptake [5]. By gathering insights from PHU nurses, school staff and parents who did not consent to vaccination, the study aims to comprehensively understand the factors influencing participation and identify practical strategies to enhance consent rates and increase vaccine uptake among adolescents. The inclusion of non-consenting parents provides a unique contribution to existing research, offering a more complete picture of stakeholder perspectives. The findings from this study will inform targeted interventions, contributing to ensuring equitable access to vaccination and supporting Australia’s broader public health goals.

## 2. Materials and Methods

### 2.1. Setting

The SESLHD is a major metropolitan health district in NSW, representing a diverse population of approximately 13% of the state. Detailed demographic characteristics have been described in prior work [5]. The SESLHD PHU partners with 92 secondary schools across the district including government, independent and Catholic secondary schools [13]. In Australia, government schools are predominantly government-funded and provide secular education. Catholic and independent schools often have religious affiliations, are partially government funded and typically charge tuition fees [14,15].

### 2.2. Participants and Recruitment

This qualitative study employed a research approach involving semi-structured interviews and a focus group to gather insights from key stakeholders. The three key participant groups targeted included:

PHU nurses: A focus group was conducted with five senior staff members from the PHU, including the Immunization Manager, three nurse team leaders and the Administration Coordinator.

School staff: Semi-structured interviews were carried out with the designated school staff member responsible for administering the school vaccination program. Schools were purposively selected to reflect varying vaccination coverage levels, sectors (government, Catholic and independent) and demographic profiles. Selection was also informed by feedback from PHU nurses involved in the program’s implementation. Principals were first contacted with an introductory letter and information outlining the study. Once approval was granted, follow-up communication was made with the school’s vaccination program coordinator. Interviews were scheduled at participants’ convenience, either online or face-to-face. The aim was to recruit participants for the study, with participants receiving a AUD $50 gift card as compensation for their time. A total of 28 schools were approached to achieve the recruitment goal. Schools that did not participate were not able to accommodate the research schedule or did not respond to the invitation.

Parents: Semi-structured interviews were conducted with parents of students who were not fully vaccinated at school. Schools assisted with recruitment by distributing invitation templates with a link to an online research page hosted on REDCap. This page contained study information, an electronic consent form and section for participant contact details. Recruitment occurred via email, with some schools using text messages to increase reach. Schools were also encouraged to send reminders to increase participation rates. Parents who expressed interest completed the consent form and indicated their availability for an interview. The research team then scheduled and conducted telephone interviews with parents at mutually convenient times. The aim was to interview 20 parents, with each participant receiving an AUD $50 gift card for their time. Schools received an AUD $150 gift card for assisting with parent recruitment.

### 2.3. Data Collection

Data collection occurred between July 2022 and January 2024, using a topic guide with open-ended questions tailored to each participant group. The interview guides were developed by L.M. and reviewed and edited by the research team to ensure thoroughness.

Interviews with school staff and PHU nurses explored key factors affecting the implementation and uptake of the school vaccination program and ongoing challenges. The focus group with PHU nurses, while limited to five participants, represented the complete senior immunization team responsible for program implementation across the district. Discussions with parents explored their general attitudes toward vaccination, experiences with the school vaccination program and barriers to participation. All groups were invited to suggest strategies for improving program delivery and uptake.

L.M. and A.Y. conducted the interviews with school staff and parents, while C.D. led the focus group with PHU nurses. L.M. and A.Y. were not previously known to the parents or school staff and C.D. had no prior relationships with PHU nurses. This dynamic helped minimize potential bias during data collection, a factor considered during analysis.

The focus group lasted approximately 86 min, while individual interviews ranged from 14 to 57 min, with an average duration of 35 min. All interviews were audio-recorded and transcribed verbatim. Written consent was obtained from all participants, and confidentiality was maintained through the secure storage of recordings and transcripts.

### 2.4. Data Analysis

Thematic analysis was performed, using both inductive and deductive approaches [16,17]. NVivo 12 software facilitated data coding. The initial coding framework was developed by C.D. and refined by L.M. and A.Y., with input from the research team to ensure alignment with study objective. Data analysis involved repeated transcript readings, coding for key themes and pattern interpretation. Key quotations illustrating identified themes were extracted and compiled, with anonymized participant identifiers used to maintain confidentiality.

Data saturation was monitored throughout the data collection and analysis process. No new themes or significant insights emerged after the interviews with the final participants, indicating that further data collection would not substantially alter the findings. This conclusion was confirmed through ongoing discussions within the research team, where it was noted that additional interviews did not yield new information, supporting the determination that data saturation had been reached.

## 3. Results

### 3.1. Participants Characteristics

A focus group discussion was conducted with five staff members from the PHU most closely involved in the school program. This group comprised the Immunization Manager, three Nursing Team Leaders and one Administrative Officer.

Semi-structured interviews were conducted with 11 school staff across seven government schools, two independent schools and two Catholic schools. Of these, eight interviews were conducted online and three were face-to-face. Participants included staff in teaching, administrative and executive roles. Some interviews were conducted before and some after the implementation of the online consent system (Appendix A).

Eleven interviews were conducted with mothers of students who were not fully vaccinated at school. Data collection consisted of 10 telephone interviews and one written response. The parents varied in educational background, with most holding university-level qualifications. The adolescents of these parents included six females and five males. In terms of vaccination status, five adolescents were unvaccinated, three were partially vaccinated with the dTpa vaccine and three were fully vaccinated by their local physician (Appendix A).

Of the unvaccinated adolescents, three parents preferred physician-administered vaccination but had not yet followed through with this plan. Another parent had provided consent, but their child was absent on the school clinic day and refused to attend the catch-up clinic. The parent intended to take the child to a physician for vaccination. One parent explicitly refused vaccination for their adolescent.

Findings revealed three distinct groups of parents who did not provide consent for their adolescents to receive vaccinations through the school-based immunization program: (1) those preferring physician-administered vaccines, (2) the accidentally non-consenting and (3) the vaccine hesitant. Additionally, stakeholders proposed strategies to improve consent and uptake (Table 1).

### 3.2. Preference for Physician-Administered Vaccines

A notable trend described by parents and school staff in the study was the preference for students to receive vaccinations at their general practitioner (GP) rather than in the school setting. This preference was driven by three main factors.

#### 3.2.1. Trust, Familiarity and Comfort

Many parents valued the long-standing relationships with their GPs, perceiving them as more reliable and personal. One parent expressed:

“*Because we’ve had the same GP like forever. So, I trust him and he’s been [my son’s] GP from birth. And he’s done everything*.”(Parent 4)

Concerns about side effects also played a role, with one parent noting:

“*She faints when she has, or she gets dizzy when she has injections at the GP. She didn’t respond particularly well to the COVID vaccine and so that’s why we decided not to have her injected at school in that environment just for safety really … The GP was prepped and ready for her knowing that she’d had the reaction prior. … but the next round they’re going to get at school, is anyone going to notice? Are the staff going to be across that?*”(Parent 3)

Additionally, parents believed their child would be more comfortable receiving vaccinations in a familiar environment, as noted by another parent:

“*She just asked could I be there. That’s all. She just wanted me to be there with her to do it*.”(Parent 1)

#### 3.2.2. Seeking Privacy and Avoidance of Social Discomfort

Both parents and school staff noted students’ concerns about receiving vaccines in front of peers. One school member observed:

“*I think it’s more just an anxious thing that they want to withdraw and go to the GP so that they’re not getting a vaccination in front of everyone*.”(School staff 11)

A parent highlighted the influence of peer dynamics:

“*they were really nervous with having it done with everyone and everyone watching because they often cry and things like that. And to do that in front of all your mates in high school; not a good look*.”(Parent 11)

Another parent emphasized the comfort provided by the GP setting:

“*you’re going to a space where you feel safe with your GP. You don’t have your friends in front of you laughing or carrying on or whatever. It’s just a safe environment that you trust and you know*.”(Parent 4)

Concerns about privacy during school vaccinations were also noted, with one parent stating:

“*Just the privacy. So, I think having more privacy when she was taken in to get the shot would’ve been better*.”(Parent 8)

#### 3.2.3. Perceived Convenience

Parents valued the flexibility and control over the timing of vaccinations offered by the GP setting, allowing them to avoid conflicts with other activities or health issues, one parent explained:

“*We didn’t get it done over summer because he does a lot of cricket and he does rep cricket and I was worried, with the tetanus, it was going to hurt his arm a bit and I know that he would be stressed about that. And he doesn’t like anything that upsets his cricket game so that’s why we were waiting right to the end of the cricket season to have it done*.”(Parent 2)

### 3.3. “Accidentally Non-Consenting”: Logistical and Operational Challenges

A substantial number of cases of non-consent were unintentional, resulting from various logistical and operational barriers. These challenges were categorized into three main areas.

#### 3.3.1. Administrative and Communication Barriers

Despite efforts to communicate effectively, many families engaged with the consent process only at the last minute or not at all, causing difficulties for school staff. As one staff member noted:

“*What I tend to find in my school community is that people tend to do things at the last minute, so I’m usually getting a lot of calls and emails the morning of vaccinations*.”(School staff 11)

Additionally, some parents did not thoroughly read the provided information, as another staff member noted:

“*The majority don’t, I would say. That’s from how many email responses I get back from parents*.”(School staff 3)

Inadequate communication about catch-up clinics also contributed to missed vaccinations, with one parent mentioning: “*Probably the GP. I’m not sure if the school offers it again*.” (Parent 8).

#### 3.3.2. Language Barriers

For families with limited English proficiency, completing consent forms presented a significant challenge. One staff member highlighted the issue:

“*Sometimes our kids who have English as a second language or parents who don’t speak English at home struggle to get the forms filled out correctly*.”(School staff 3)

Students often helped bridge this gap by translating for their parents, as described by another staff member:

“*So we do have a lot of parents who speak another language. Often if we’re vaccinating and the kids are getting their second injection or they missed the vaccination and we have to catch them up and I wasn’t able to speak to mum, I get the kids to call mum or dad and they speak in their language … they get on the phone, they can translate to the parents, ‘mum, we’re getting vaccinated do you consent?’*.”(School staff 2)

#### 3.3.3. Socioeconomic Factors and Attendance Issues

While the free nature of the school vaccination program helps mitigate some barriers, students from lower socioeconomic backgrounds or challenging family situations often faced difficulties A staff member explained:

“*I’d probably say some students that are low socioeconomic status struggle sometimes. Or students sometimes with wellbeing issues from difficult family backgrounds, maybe challenging parents or split parents sometimes can struggle too*.”(School staff 1)

Those with chronic poor attendance also faced additional barriers to participation:

“*So, if there’s students with low attendance, sometimes they struggle to get it back as well because they miss messages. Sometimes they think they’ve handed it in, but they haven’t."*(School staff 1)

This was echoed by a focus group participant:

“*There is one school which we do know generally has low uptake and we do know that perhaps the compliance of these students returning the card could be because they just don’t—it’s not on their priority, these students wouldn’t return it to their parents. They have poor absence, they’re not at school nine times out of 10, so you just know they’re not even going to be there on the day that we’re there for the vaccination*.”(Focus group)

### 3.4. Vaccine Hesitancy

While representing a smaller group of non-consenting parents, vaccine hesitancy, particularly regarding the HPV vaccine, emerged as a significant theme. Several interconnected factors contributed to this hesitancy.

#### 3.4.1. Safety Concerns and Misinformation

Parents expressed concerns about the long-term effects of the HPV vaccine, perceiving it as newer compared to established vaccines like dTpa. One parent noted:

“*I just don’t know enough about the HPV injection, the long-term effects of it, because it hasn’t been around for that long to be able to see what’s going to happen when she turns 30, will this impact her in anyway? There hasn’t been any—I don’t even know if it’s been out long enough for that yet to happen*.”(Parent 9)

Misinformation about infertility, cancer risks and other side effects also contributed to hesitancy, as seen in statements like:

“*It just has come up and it can actually cause cervical cancer*.”(Parent 1)

Stories about adverse reactions further fueled these concerns, with one parent questioning the vaccine’s role in their family member’s “*fertility issues*” (Parent 9).

#### 3.4.2. Cultural and Religious Beliefs

Cultural and religious values also influenced vaccine acceptance. Some parents viewed the HPV vaccine as promoting early sexual activity, which conflicted with their beliefs. One parent commented:

“*I don’t give consent for my kids to be sexually active. If you only have sex with one or two, the problem won’t be there. The problem, the sickness, will be there if you have multiple sex or—you know… Sex is supposed to be marriage. Obviously, it’s not always the case but I don’t want you to—changing partner and have sex all the time*.”(Parent 5)

Others preferred to delay the vaccine, considering it unnecessary until their children were older:

“*It was more the sexual activities that we were thinking about and thinking, oh, she doesn’t need it just yet and we didn’t want to put her through that stress*.”(Parent 8)

Cultural and religious backgrounds also shaped vaccination attitudes for some families. A school staff member affirmed:

“*It’s actually a cultural thing because we have an Islamic community. And I’m Islamic myself, so I understand why the parents have chosen not to go with HPV. It’s the sexually transmitted disease one that they often don’t want injected into the body*.”(School staff 2)

#### 3.4.3. Risk Perception

Perceptions of risk played a role in hesitancy, with some parents viewing the HPV vaccine’s long-term risks as less immediate compared to vaccines like dTpa. For example, one parent noted:

“*I think it’s because of her age, you don’t think about the cancers and the sexual activity just yet. But the whooping cough is viral, you could just catch it from anywhere, going to shopping centers. Tetanus, she could step onto a rusty nail*.”(Parent 8)

Gender-specific risk perceptions also played a role:

“*I have decided that since my boys don’t have a cervix—I don’t think they have a chance of developing cervical cancer—however it does appear that there is a larger risk of neurological issues developing in the years after receiving this product*.”(Parent 10)

#### 3.4.4. Impact of COVID-19

The COVID-19 pandemic exacerbated vaccine hesitancy, increasing anxiety and mistrust in health authorities. School staff observed:

“*We have had unfortunately, some parents just going through COVID, have turned into vaccination experts*.”(School staff 2)

The pandemic’s impact on vaccine attitudes was evident in parents’ comments:

“*I think because of the coercion and the bullying we felt through COVID-19, it made people either fold and get it, or stand stronger and say no. And that’s when I think we got to that point, we stood stronger and just went No*.”(Parent 9)

The abundance of conflicting information during the pandemic further complicated decision-making:

“*It’s really difficult to make the right decision because you can’t even believe the health anymore because of all the controversy going on…and it’s just so much information*.”(Parent 8)

### 3.5. Stakeholder Strategies and Recommendations

Stakeholders proposed a multi-faceted approach to improving vaccination uptake, emphasizing the need for clear, accessible information delivered through various channels to both parents and students. These strategies aim to address knowledge gaps, combat misinformation and alleviate concerns about the vaccination process. The involvement of trusted healthcare providers and targeted educational efforts were highlighted as key elements in overcoming vaccine hesitancy and improving overall vaccination rates.

#### 3.5.1. Enhanced Communication and Educational Strategies

Many stakeholders recommended robust communication campaigns using various media platforms to educate parents and students about vaccine benefits and safety. A school staff member suggested:

“*Everyone watches TV or they’re on social media, so I just think that little pop-up will kind of just trigger their memory*.”(School staff 7)

Another proposed leveraging platforms popular among youth:

“*They should put something on social media, like put a TikTok out, a TikTok dance about vaccinations, right, because you will hit 99% of kids that way*.”(School staff 4)

Additionally, stakeholders emphasized the need for language-accessible materials. As one school staff member noted:

“*Whether it’s a PowerPoint or it’s a short link video or something that can explain what the vaccination is for, what it prevents and how it’s beneficial for them and the community. I think that would be a really great tool, especially for the parents who are newly arrived to Australia and often speak another language*.”(School staff 2)

#### 3.5.2. Addressing Student Anxiety and Awareness

Educational efforts targeting students were highlighted as crucial for reducing anxiety and improving understanding. One school staff member described their approach:

“*I just made really interesting PowerPoint slides that go on for 10 min, like what is meningococcal disease, why do you need to get vaccinated against it, what are the symptoms and so on… Because I feel like because we give the consent form to the parents, the kids kind of just feel a bit lost*.”(School staff 10)

Parents also emphasized the need for clear communication with students. One parent suggested:

“*Speak to the kids about it. Just to calm their nerves or just, make them know what they’re actually getting because when he said, oh, it’s the HIV, I’m like, oh, he doesn’t know; he doesn’t know*.”(Parent 7)

Parents highlighted the importance of clear explanations and privacy during the vaccination process:

“*Maybe if they explained it a bit more to the kids and—because I’m not even sure what they’re doing the way they’re doing it at school or whether they just line them up and jab them all in a row or whether they individually take them in a room and do it in a room. But maybe if that was explained to them and they were individually doing it, he might not be as anxious about it*.”(Parent 2)

“*Because of how anxious she is, I think it would’ve been nicer to have a bit more privacy while getting the injection so that it didn’t add extra embarrassment to the process*.”(Parent 8)

#### 3.5.3. Parent Education and Engagement

Stakeholders suggested engaging parents through informational events, such as information nights or live streams. As one parent proposed:

“*Having an information night and we’ll zoom, we’ll do a live stream and you can all come along, bring your kids*.”(Parent 3)

Integrating vaccination information into existing school events was also recommended:

“*A lot of the information is there, it is just getting them to read it, but if you’ve got the forum, especially with Year 7 for their information night*.”(School staff 8)

Addressing cultural and religious concerns were identified as crucial, with one school staff member suggesting targeted parent programs:

“*If they’re not consenting to both, it’s usually the Islamic kids who are not taking the HPV one, that would be the only reason. It probably could be good to have a parent program around that. And if there’s any myths or if there’s any benefits and it becomes quite clear to the community what that could be*.”(School staff 2)

Stakeholders also emphasized the importance of addressing knowledge gaps and misconceptions. As one focus group participant noted:

“*Getting questions like, ‘Are boys supposed to be getting the HPV vaccine, or is that supposed to be just for the girls?’ From some of the principals as well*.”(Focus group)

To combat misinformation, partnering with primary health networks and leveraging trusted healthcare providers was suggested:

“*Would [GPs] work with us to try and develop some resources that could be handed out, because I think parents have got a trusted relationship with their GP and so they might be able to have that open conversation with their GP, about their hesitancy and things like that*.”(Focus group)

Additionally, one parent highlighted the potential for proactive reminders from health authorities:

“*Like even a letter from the vaccination registry or immunization registry that, you know, you still haven’t done—so [my son’s] 13 now, so that would have prompted me as well to go and do it and I haven’t got anything like that*.”.(Parent 11)

## 4. Discussion

This study identifies key factors influencing vaccination uptake in school-based vaccination programs, emphasizing vaccine hesitancy, preferences for GP-administered vaccines and logistical barriers. Our findings advocate for tailored interventions that recognize the diverse reasons behind vaccination uptake, which are essential for enhancing vaccination rates and improving program efficacy.

### 4.1. Barriers to Participation

The preference for GP-administered vaccines reflects a broader reliance on trusted healthcare providers, highlighting the role of established relationships in promoting vaccine confidence [18,19]. However, the “intention–behavior gap” observed, where only three out of seven parents intending to vaccinate through GPs followed through, underscores the need for system-level interventions, such as integrated follow-ups and reminders to bridge the gap between intention and action [20]. Social factors, such as peer pressure and privacy concerns, also influenced decisions. Fear of embarrassment in front of peers was a common reason for preferring GP settings, suggesting that adolescents’ sensitivity to peer perceptions could be addressed in school-based vaccination interventions [21,22].

Accidental non-consent further limited participation. Parents cited logistical challenges, including not receiving or understanding the forms, forgetting to return them, or miscommunication between schools and parents. These procedural barriers, rather than deliberate refusal or vaccine hesitancy, are consistent with other research identifying logistical issues as significant vaccination barriers [23]. Language barriers further exacerbated these challenges. While translated materials were available, their distribution often failed to reach intended audiences effectively and the shift to online consent further complicated matters for non-English-speaking families, removing opportunities for students to assist their parents. These findings align with prior research indicating that language barriers significantly affect vaccination rates among culturally and linguistically diverse populations, often leading to misunderstandings and difficulties in completing consent forms correctly [8]. Additionally, families with limited digital literacy or unreliable internet access struggled, consistent with findings on the digital divide in health services, where lower digital literacy hinders access to online health resources [24]. Ensuring that consent materials are accessible and comprehensible to all families is crucial [23,24].

Vaccine hesitancy emerged as a central challenge, with parental concerns often stemming from misinformation, safety apprehensions and cultural or religious beliefs. For instance, apprehension about the long-term effects of the HPV vaccine, consistent with previous studies that report similar doubts despite extensive evidence supporting its benefits and long-term safety [25,26]. Some concerns were linked to the vaccine’s association with sexual activity, reflecting moral or religious values, consistent with other studies [9,27,28].

The COVID-19 pandemic exacerbated these issues, amplifying vaccine-related anxieties and shaping parental attitudes not only toward COVID-19 vaccines but also other routine vaccinations. This aligns with recent research, which suggests that public discourse around COVID-19 vaccines has shaped perceptions of routine immunization programs [29]. The pandemic has amplified existing concerns and introduced new ones, altering how individuals weigh the risks and benefits of vaccination. These shifts highlight the need for targeted communication strategies that address both longstanding and emerging concerns about vaccine safety and efficacy.

### 4.2. Addressing Barriers and Policy Implications

Both the study results and literature emphasize that clear, evidence-based communication is crucial to counter misinformation and address specific fears surrounding vaccines [11,30]. Stakeholders highlighted the need to use relatable media and diverse channels, such as social media platforms popular with parents and students, aligning with research on multimedia interventions’ impact on health behaviors [31,32]. Additionally, organizing information nights or live-streamed sessions led by school nurses was recommended to build trust and offer real-time answers, an effective method for tackling vaccine hesitancy [33,34].

Culturally sensitive communication and multilingual educational materials are essential for improving vaccine acceptance [28]. Engaging leaders who share cultural or religious backgrounds with hesitant parents is a key strategy for addressing community-specific concerns, such as the role of vaccines in disease prevention, rather than fears around sexual activity or gender-specific perceptions of vaccination. These leaders can help bridge trust gaps, address concerns and encourage vaccine uptake [35].

Recent evidence from Italy demonstrates the effectiveness of network-based approaches in improving HPV vaccination coverage. Guarducci et al. (2024) found that developing integrated networks between healthcare providers, schools and local communities significantly improved vaccination rates through better coordination and resource sharing [36]. Their model, which emphasized systematic communication channels between stakeholders, could be adapted to address the intention–behavior gap identified in our study, particularly in coordinating follow-up between schools and healthcare providers.

Addressing student anxiety is crucial for improving school-based immunization programs. Stakeholders emphasized the need for more adolescent-specific information, echoing concerns from a NSW study where both parents and students’ concerns and misconceptions, such as cancer being contagious, were raised [27]. Engaging educational content, like pamphlets or TikTok videos, can help students better understand vaccines, making them more likely to advocate for their health and positively influence parental decisions [31,32,33,37]. Furthermore, innovative digital interventions show promise in addressing vaccine hesitancy. A recent study in South Korea demonstrated that a six-week smartphone-based educational program significantly improved mothers’ knowledge and attitudes toward HPV vaccination for their sons [38]. Their successful use of mobile technology to deliver targeted educational content suggests that similar digital approaches could be effective in reaching parents who prefer accessing health information through digital platforms.

Improving procedural aspects, such as ensuring privacy during school vaccinations, can address social barriers like embarrassment. Measures include optimizing the flow of students through vaccination stations and using privacy screens. Clear communication about how vaccinations are administered—whether in private or group settings—may further alleviate concerns. These interventions align with evidence that supportive environments encourage vaccine participation [22,39,40,41].

Fostering closer collaboration between schools and local GPs offers an opportunity to leverage existing trust in healthcare providers. GPs can support school-based programs by reassuring parents about vaccine safety and efficacy while facilitating follow-up for missed vaccinations. This partnership can also address the intention–behavior gap by providing reminders and coordinating catch-up campaigns, aligning with research on healthcare provider recommendations’ positive impact on vaccine acceptance [18,39,40].

Logistical barriers can be mitigated through flexible, streamlined consent processes. Combining electronic and paper options, providing multilingual support and offering on-site assistance for digital consent forms can enhance accessibility. Reminder–recall systems that use multiple communication channels and personalized messaging have been shown to improve vaccine uptake and should be prioritized [23,41].

### 4.3. Future Directions and Limitations

This study emphasizes the need for multifaceted interventions to improve vaccination uptake. Combining broad public health campaigns with targeted strategies that address specific parental concerns, logistical challenges and social influences is crucial for enhancing the effectiveness of school-based vaccination programs. Future research should evaluate the impact of these tailored interventions across diverse populations to refine best practices and ensure their applicability in varying contexts. Recent studies highlight the potential of both network-based organizational models and digital interventions in improving vaccination uptake, suggesting that these approaches warrant further investigation in the Australian context [36,38]. Long-term investigations are also needed to understand the pandemic’s effects on vaccine perceptions and school-based vaccination programs. Insights from these studies could inform strategies for addressing hesitancy in a post-pandemic landscape.

The study’s strengths include its in-depth exploration of barriers from multiple perspectives, including those of parents who did not consent to vaccination—a group often underrepresented in vaccine hesitancy research. This study specifically targeted understanding the barriers and challenges faced by parents who declined or were indecisive about school-based vaccination programs. While existing literature primarily focuses on the perspectives of parents who consent to these programs, our research complements this body of work by providing a detailed exploration of the obstacles and concerns preventing participation. However, future research could aim to synthesize findings from both consenting and non-consenting groups to provide a more holistic view of factors influencing vaccine participation.

Although the qualitative design inherently involves smaller sample sizes compared to quantitative research, the purposive recruitment of non-consenting parents, a hard-to-reach population, enhances the study’s depth and relevance. The inclusion of these individuals allowed for a rich understanding of their unique barriers and concerns, which might otherwise remain overlooked. However, several limitations should be acknowledged.

The study’s focus on the South Eastern Sydney Local Health District, a major metropolitan area in NSW, may limit the generalizability of findings to regions with different demographics or socio-economic characteristics. While the study provides valuable insights into the barriers faced by non-consenting parents in this specific context, the findings may not fully reflect the challenges experienced in other areas, particularly rural or regional areas, or in countries with different healthcare infrastructures.

We acknowledge that social desirability bias may have influenced participants’ responses, especially given the sensitive nature of vaccination decisions in the post-COVID-19 context. However, several aspects of our methodology helped mitigate this concern. First, our recruitment of parents who did not consent to vaccination and their willingness to express negative views suggests that participants felt comfortable sharing their opinions. Second, the semi-structured interview format encouraged open-ended discussions, allowing participants to freely express their hesitations and concerns. Finally, interviews were conducted by researchers not involved in the vaccination program, minimizing pressure to provide ‘acceptable’ answers. Nevertheless, some participants may have moderated their views, particularly on more controversial aspects of vaccine hesitancy.

Recruitment challenges, such as reliance on school staff to identify non-consenting parents, led to lower-than-anticipated participation rates, potentially affecting the breadth of perspectives gathered. Additionally, the lack of adolescent perspectives limits the study’s scope, as understanding the motivations and concerns of students could provide a more holistic view of vaccine uptake.

To enhance the applicability of these findings, future research should replicate this study design across diverse geographical and socioeconomic contexts, including rural and remote areas, different healthcare systems and varying cultural settings. Additionally, integrating adolescents’ views would provide a more comprehensive understanding of their motivations and concerns regarding vaccinations in school settings.

## 5. Conclusions

This qualitative study offers valuable insights into the multifaceted factors influencing school-based vaccination programs, emphasizing the need to address vaccine hesitancy, preferences for GP-administered vaccines and logistical barriers. While findings are derived from a metropolitan health district in NSW, the identified challenges and solutions contribute to our understanding of barriers facing school vaccination programs in comparable urban settings.

The proposed strategies, ranging from enhanced communication and digital interventions to streamlined consent processes and healthcare provider partnerships, offer a comprehensive approach to improving program accessibility and coverage. Recent evidence supporting network-based approaches and smartphone-based educational programs demonstrates the potential effectiveness of such interventions. Implementing these measures may require coordinated policy changes at multiple levels, from school administration to health departments. Addressing these diverse challenges has the potential to significantly enhance the effectiveness, equity and reach of school-based vaccination programs, contributing to improved public health outcomes.

## Figures and Tables

**Table 1 vaccines-13-00083-t001:** Summary of themes and sub-themes.

Theme	Key Factors
1. Preference for physician-administered vaccines	Trust in physician, personalized care, familiarity with provider, seeking privacy and avoidance of peer pressure, perceived convenience
2. Accidental non-consent	Miscommunication, logistical issues, form distribution, language barriers, socioeconomic factors and attendance issues
3. Vaccine hesitancy	Safety concerns, misinformation, lack of education, cultural and religious beliefs, risk perception, COVID exacerbating concerns
4. Stakeholder strategies and recommendations	Enhanced communication and education, addressing student anxiety and awareness, parent education and engagement

## Data Availability

The original contributions presented in the study are included in the article, further inquiries can be directed to the corresponding authors.

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
