# Peer review of "Provider Preference, Logistical Challenges, or Vaccine Hesitancy? Analyzing Parental Decision-Making in School Vaccination Programs: A Qualitative Study in Sydney, Australia"

_vaccines, 2025, doi:10.3390/vaccines13010083_

Round 1

Reviewer 1 Report

Comments and Suggestions for Authors

Dear Authors,

The article is interesting, below my comments:

-Line 39-49: I suggest improving this part, is important for the reader understand the context better.

- The discussion should be improved, for this I suggest the following article:

Guarducci G, Chiti M, Fattore DC, Caldararo R, Messina G, Filidei P, Nante N. Development of a network model to implement the HPV vaccination coverage. Ann Ig. 2024 Nov-Dec;36(6):636-643. doi: 10.7416/ai.2024.2630. 

Cho Y-H, Kim T-I. A Six-Week Smartphone-Based Program for HPV Prevention Among Mothers of School-Aged Boys: A Quasi-Experimental Study in South Korea. Healthcare. 2024; 12(23):2460. https://doi.org/10.3390/healthcare12232460

Author Response

Comment 1: The article is interesting, below my comments:

Line 39-49: I suggest improving this part, is important for the reader understand the context better.

Response 1: We appreciate your suggestion to improve the context for better reader understanding. In response, we have revised this section to provide a clearer and more comprehensive explanation of the National Immunisation Program (NIP) and the specific vaccines offered in the school-based vaccination program. We have ensured that the historical context and key details are presented more cohesively to enhance the reader's understanding of the topic.

Comment 2: The discussion should be improved, for this I suggest the following article:

Guarducci G, Chiti M, Fattore DC, Caldararo R, Messina G, Filidei P, Nante N. Development of a network model to implement the HPV vaccination coverage. Ann Ig. 2024 Nov-Dec;36(6):636-643. doi: 10.7416/ai.2024.2630. 

Cho Y-H, Kim T-I. A Six-Week Smartphone-Based Program for HPV Prevention Among Mothers of School-Aged Boys: A Quasi-Experimental Study in South Korea. Healthcare. 2024; 12(23):2460. https://doi.org/10.3390/healthcare12232460

Response 2: Thank you for suggesting these additional references to improve the discussion section. In response to your comment, we have reviewed the suggested articles and incorporated relevant findings into the discussion to strengthen the connections between our study and the broader body of literature on HPV vaccination.

Reviewer 2 Report

Comments and Suggestions for Authors

The research focus factors influencing the implementation of school-based immunization programs. The authors investigates patterns of vaccine decision-making within Australia's school-based immunization program. The study used semi-structured interviews to investigate three groups of people. Research shows that there are many reasons for not participating in school vaccination programs. Stakeholder should emphasized tailored strategies such as strengthening GP collaboration for parents preferring physician settings, streamlining consent processes for those facing logistical barriers. This study provides feasible solutions for the social determinants of vaccine acceptance. Expected to help improve vaccination rates and address indecisiveness in school programs.

The author seems to have written the wrong title for 3.5. It should be 'Stakeholder strategies and recommendations', however it is "“Accidentally non-consenting”: Logistical and operational challenges".The representative population of the study is relatively limited. Only applicable to a major metropolitan health district in NSW, with limited reference value for populations in other parts of the world. The depth of exploration of the results in the entire study is also insufficient.

Author Response

Comment 1: The research focus factors influencing the implementation of school-based immunization programs. The authors investigates patterns of vaccine decision-making within Australia's school-based immunization program. The study used semi-structured interviews to investigate three groups of people. Research shows that there are many reasons for not participating in school vaccination programs. Stakeholder should emphasized tailored strategies such as strengthening GP collaboration for parents preferring physician settings, streamlining consent processes for those facing logistical barriers. This study provides feasible solutions for the social determinants of vaccine acceptance. Expected to help improve vaccination rates and address indecisiveness in school programs.

The author seems to have written the wrong title for 3.5. It should be 'Stakeholder strategies and recommendations', however it is "“Accidentally non-consenting”: Logistical and operational challenges".

Response 1: Thank you for pointing out the discrepancy in the title of section 3.5. We apologize for the oversight.  We have made the necessary correction in the manuscript, and the section is now titled as per your suggestion.

Comment 2: The representative population of the study is relatively limited. Only applicable to a major metropolitan health district in NSW, with limited reference value for populations in other parts of the world. 

Response 2: Thank you for your careful review and feedback regarding the study's geographical limitations. While we had acknowledged the geographic constraints in our original manuscript, we agree that this warranted more detailed discussion. We have therefore enhanced the limitations section to more explicitly address the specific characteristics of our study setting within the South Eastern Sydney Local Health District. While the findings may be context-specific, they still provide valuable insights that could inform further studies in different areas, especially when considering the potential transferability of strategies for addressing vaccine uptake barriers across diverse regions and healthcare settings.

Comment 3: The depth of exploration of the results in the entire study is also insufficient.

Response 3: We respectfully note that while deeper analysis is always valuable, two other peer reviewers found the current depth of analysis satisfactory for addressing our research aims and contributing meaningfully to the field. Nevertheless, we appreciate your perspective.

To ensure we have not overlooked opportunities for deeper analysis:

  1. We have strengthened our discussion by incorporating recent literature, including the works by Guarducci et al. (2024) and Cho & Kim (2024), which provide additional context and support for our findings
  2. Our analysis includes rich qualitative data from typically hard-to-reach participants (non-consenting parents), offering unique insights into vaccination barriers
  3. The study triangulates perspectives from multiple stakeholders (PHU nurses, school staff, and parents) to provide a comprehensive understanding of the issues

If you have specific aspects of the results that you feel warrant deeper exploration, we would welcome your detailed suggestions to help us further enhance the manuscript's contribution to the field.

Reviewer 3 Report

Comments and Suggestions for Authors

Thank you for the opportunity to review this manuscript. I found it very interesting and appreciated the comprehensive insights into school-based vaccination. However, I have a few concerns that I believe should be addressed. Please see my comments below:

#1. Study title:

This study is a qualitative study, which is a key feature of the research. However, this is not reflected in the study title. Including the term "qualitative study" in the title would be beneficial for readers.

#2. Social desirability bias:

Participants may have provided socially desirable answers, potentially obscuring their true motivations and opinions. It would be valuable to include a discussion on this point.

#3. Lack of perspectives from consenting parents:

The study does not include the perspectives of parents who consented to school-based vaccinations, which limits the understanding of the overall situation. Addressing this limitation in the discussion would strengthen the study.

#4. Data saturation:

It is unclear whether data saturation was achieved. Demonstrating that data saturation was reached (e.g., by documenting when no new themes emerged) is crucial to ensuring that the data collection covered diverse themes and perspectives. Providing evidence of data saturation would enhance the value of the study.

Author Response

Comment 1: Thank you for the opportunity to review this manuscript. I found it very interesting and appreciated the comprehensive insights into school-based vaccination. However, I have a few concerns that I believe should be addressed. Please see my comments below:

#1. Study title:

This study is a qualitative study, which is a key feature of the research. However, this is not reflected in the study title. Including the term "qualitative study" in the title would be beneficial for readers.

Response 1: Thank you for highlighting the importance of accurately reflecting the qualitative nature of the study in the title. We have revised the title to explicitly include "qualitative study," ensuring that readers can immediately recognise the methodological approach of the research.

Comment 2: #2. Social desirability bias:

Participants may have provided socially desirable answers, potentially obscuring their true motivations and opinions. It would be valuable to include a discussion on this point.

Response 2: Thank you for bringing this point to our attention. We agree that this merits discussion and have added e following paragraph acknowledging social desirability to our limitations section:

“We acknowledge that social desirability bias may have influenced participants' responses, especially given the sensitive nature of vaccination decisions in the post-COVID-19 context. However, several aspects of our methodology helped mitigate this concern. First, our recruitment of parents who did not consent to vaccination and their willingness to express negative views suggests participants felt comfortable sharing their opinions. Second, the semi-structured interview format encouraged open-ended discussions, allowing participants to freely express their hesitations and concerns. Finally, interviews were conducted by researchers not involved in the vaccination program, minimizing pressure to provide 'acceptable' answers. Nevertheless, some participants may have moderated their views, particularly on more controversial aspects of vaccine hesitancy.”

Comment 3: #3. Lack of perspectives from consenting parents:

The study does not include the perspectives of parents who consented to school-based vaccinations, which limits the understanding of the overall situation. Addressing this limitation in the discussion would strengthen the study.

Response 3: Thank you for your insightful feedback. We acknowledge that including perspectives from consenting parents would provide a more comprehensive understanding of the factors influencing school-based vaccination programs. However, the focus of our study was specifically on barriers to participation, targeting parents who declined or were indecisive about school-based vaccinations. This focus allows us to identify challenges and actionable strategies for improving vaccine uptake among hesitant or non-consenting populations.

We have revised the discussion to explicitly acknowledge the broader literature on consenting parents and positioned our study as complementary to these works. Additionally, we highlighted the potential for future research to synthesise findings from both consenting and non-consenting groups to present a holistic view of vaccination decision-making.

We believe this clarification strengthens the manuscript by situating our study within the broader research landscape and emphasising its unique contribution.

Comment 4: #4. Data saturation:

It is unclear whether data saturation was achieved. Demonstrating that data saturation was reached (e.g., by documenting when no new themes emerged) is crucial to ensuring that the data collection covered diverse themes and perspectives. Providing evidence of data saturation would enhance the value of the study.

Response 4: Thank you for your valuable comment regarding data saturation. We agree that explicitly documenting our data saturation approach would strengthen the manuscript

We observed that no new themes emerged after conducting interviews with the final participants from each group. This was further supported by detailed note-taking and thematic coding during the data analysis process. We have now included this information in the revised Data Collection and Analysis sections of the methods.

Round 2

Reviewer 1 Report

Comments and Suggestions for Authors

Dear Authors, for me it is ok.

Good Luck

Reviewer 3 Report

Comments and Suggestions for Authors

Thank you for addressing the comments provided during the previous round of review. The revisions and additional content have been thoughtfully made, and I find the manuscript to be significantly improved.

At this time, I do not have any additional comments or suggestions. Thank you for your careful attention to the feedback.